# STATISTICALLY CONSISTENT SALIENCY ESTIMATION

## ABSTRACT

The use of deep learning for a wide range of data problems has increased the need for understanding and diagnosing these models, and deep learning interpretation techniques have become an essential tool for data analysts. Although numerous model interpretation methods have been proposed in recent years, most of these procedures are based on heuristics with little or no theoretical guarantees. In this work, we propose a statistical framework for saliency estimation for black box computer vision models. We build a model-agnostic estimation procedure that is statistically consistent and passes the saliency checks of Adebayo et al. (2018b). Our method requires solving a linear program, whose solution can be efficiently computed in polynomial time. Through our theoretical analysis, we establish an upper bound on the number of model evaluations needed to recover the region of importance with high probability, and build a new perturbation scheme for estimation of local gradients that is shown to be more efficient than the commonly used random perturbation schemes. Validity of the new method is demonstrated through sensitivity analysis.

## 1 INTRODUCTION

Deep learning models have achieved great predictive performance in many tasks. However, these complex, often un-tractable models are difficult to interpret and understand. This lack of interpretability is a major barrier for their wide adoption, especially in domains (e.g., medicine) where models need to be qualitatively understood and/or verified for robustness.

In order to address these issues, several interpretation approaches have been proposed in the last few years. A group of methods are based on visualizations, either by quantifying the effect of particular neurons or features, or by creating new images that maximize the target score for specific classes (Erhan et al., 2009; Simonyan et al., 2013; Zeiler & Fergus, 2014). A large collection of the techniques build saliency maps by attributing the gradients of the neural network to the input image through various procedures or by finding perturbations that significantly change the output(Springenberg et al., 2014; Bach et al., 2015; Montavon et al., 2017; Shrikumar et al., 2017; Zhou et al., 2016; Selvaraju et al., 2017; Smilkov et al., 2017; Fong & Vedaldi, 2017; Adebayo et al., 2018a; Dumitru et al., 2018; Singla et al., 2019).

Another class of approaches treat the deep learner as a black-box. In this domain, Baehrens et al. (2010) use a Parzen window classifier to approximate the target classifier locally. Ribeiro et al. (2016) propose the LIME procedure, where small perturbations on the instance are used to obtain additional samples with which a sparse linear model is fit. Lundberg & Lee (2017) propose SHapley Additive exPlanation(SHAP), which combines the Shapley value from the game theory with the additive feature attribution methods. They also make connections of the SHAP procedure with various existing methods including LRP, LIME and DeepLIFT. Chen et al. (2019) propose L- and C-Shapley procedures which can reliably approximate the Shapley values in linear time with respect to the number of features.

Majority of the listed methods are heuristics which are constructed according to certain desirable qualities. For these methods, it is not clear what the main estimand is, if it can be consistently estimated or if (and how) the estimand can be computed more efficiently. In fact, according to the recent research by Adebayo et al. (2018b), most methods with great visual inspection lack sensitivity to the model and the data generating process. Theoretical explanation for why guided back-propagation and deconvolutional methods perform image recovery is provided by Nie et al. (2018).

In this work, we propose a statistically valid technique for model-agnostic saliency estimation, and prove its consistency under reasonable assumptions. Furthermore, our method passes the sanity checks given by Adebayo et al. (2018b). Through our analysis, we obtain insights into how to improve the accuracy and reliability of our approach.

We note that there is recent work by Burns et al. (2019) where they provide a saliency estimation technique with theoretical guarantees – more specifically, FDR control. Although their procedure is very promising from a statistical perspective, and theoretically valid under a very general set of assumptions, their technique requires human input and has a significant computational load as it uses a generative model for filling in certain regions of the target image.

Our main contributions are as follows:

- We introduce a new saliency estimation framework for CNNs and propose a new method based on input perturbation. Our procedure requires solving a linear program, and hence the estimates can be computed very efficiently. Furthermore, the optimization problem can be recast as a "parametric simplex" (Vanderbei, 2014), which allows the computation of the full solution path in an expedient manner.

- We establish conditions under which the significant pixels in the input can be identified with high probability. We present finite-sample convergence rates that can be used to determine the number of necessary model evaluations.

- We find that the noise distribution for the perturbation has a substantial effect on the convergence rate. We propose a new perturbation scheme which uses a highly correlated Gaussian, instead of the widely used independent Gaussian distribution.

In the following section, we define the linearly estimated gradient (LEG), which is the saliency parameter of interest (i.e. the estimand), and introduce our statistical framework. In section 3, we propose a regularized estimation procedure for LEG that penalizes the anisotropic total-variation. We provide our theoretical results in Section 4 and the result of our numerical comparisons in Section 5.

## 1.1 NOTATION

For a matrix $B$, we use $\text{vec}(B)$ and $\text{vec}^{-1}(B)$ to denote its vectorization and inverse vectorization, respectively. The transpose of a matrix $B$ is given by $B^T$ and we use $B^+$ for its pseudo-inverse . The largest and smallest eigenvalue of a symmetric matrix $B$ are denoted by $\lambda_{\max}(B)$ and $\lambda_{\min}(B)$. For a set $S$, we use $S^C$ to denote its complement. For a vector $u \in \mathbb{R}^p$ and a set $S \subseteq [1, \ldots, p]$, we use $u_S$ to refer to its components indexed by elements in $S$. The $q$-norm for a vector $u$ is given by $\|u\|_q$ and we use $\|B\|_{Fr}$ for the Frobenius norm of a matrix $B$. The vector of size $p$ whose values are all equal to 1 is denoted by $1_p$. Similarly, we use $1_{p_1 \times p_2}$ and $0_{p_1 \times p_2}$ to denote a $p_1 \times p_2$ matrix whose entries are equal to 1 and 0, respectively. Finally, for a continuous distribution $F$, we use $F + x_0$ to denote a distribution that is mean-shifted by $x_0$, i.e. $F(z) = G(z - x_0)$ for all $z$, where $G = F + x_0$.

## 2 LINEARLY ESTIMATED GRADIENT

In gradient based saliency approaches, the main goal is to recover the gradient of the deep learner with respect to the input. More specifically, let $f(x)$ be a deep learner, $f : \mathcal{X} \to [0, 1]$, where $\mathcal{X}$ is the input space, e.g., $[0, 255]^{28 \times 28}$ for the MNIST dataset, where the input are given as 28 by 28 sized images. In this notation, the output is the probability of a specific class, for instance $P_{model}(x \text{ is a } 9)$; although this can be modified to check for comparative quantities by setting the output as

$$f(x) = f_9(x) - f_7(x) = P_{model}(x \text{ is a } 9) - P_{model}(x \text{ is a } 7). \tag{1}$$

Then, local saliency is defined as the derivative of $f(\cdot)$ with respect to the input, evaluated at a point of interest $x_0 \in \mathcal{X}$, i.e. $\nabla_x f(x)|_{x=x_0}$. However, in practice, local saliency is often too noisy and one instead uses an average of the gradient around $x_0$ (Shrikumar et al., 2017; Smilkov et al., 2017).

In order to study the saliency procedure from a statistical perspective, we start by defining an estimand, whose definition is motivated by the LIME procedure (Ribeiro et al., 2016).

**Definition 1** (LEG). *For a continuous distribution $F$, an initial point $x_0 \in \mathcal{X}$ with $\mathcal{X} \subset \mathbb{R}^{p_1 \times p_2}$, and a function $f : \mathcal{X} \to [-1, 1]$, the linearly estimated gradient (LEG), $\gamma \in \mathbb{R}^{p_1 \times p_2}$ is given by*

$$\gamma(f, x_0, F) = \arg\min_g \mathbb{E}_{x \sim F + x_0} \left[ \left( f(x) - f(x_0) - vec(g)^T vec(x_0 - x) \right)^2 \right].$$

LEG is based on a first order Taylor series expansion of the function $f(x)$ around the point of interest $x_0$. The estimand is a proxy for the local gradient, and is the coefficient that gives the best linear approximation, in terms of the squared error, among all possible choices. The distribution $F$ determines the range of points the analyst wants to consider. We visually demonstrate LEG on two toy examples with a single pixel (i.e. $p_1 = p_2 = 1$) in Figure 1.

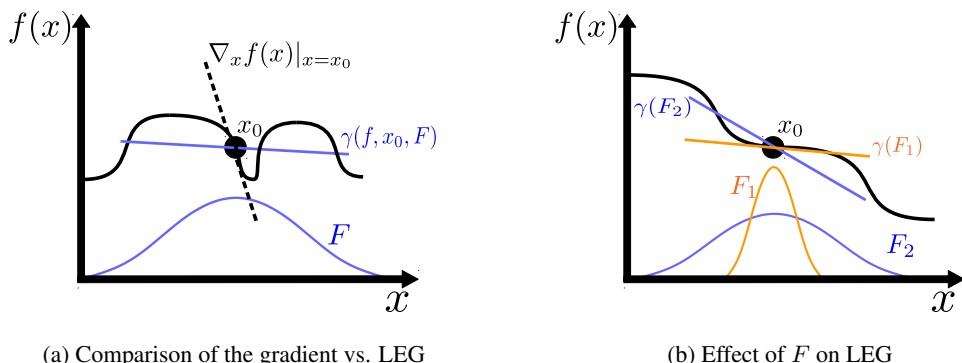

(a) Comparison of the gradient vs. LEG          (b) Effect of $F$ on LEG

Figure 1: Visual demonstrations of LEG for a single input. LEG seeks to find a local linear approximation of $f(x)$ in a neighborhood around $x_0$; choice of the distribution, $F$, determines the size of the neighborhood. In Figure 1a, we compare LEG to the gradient, which is very localized. If $f(x)$ is a highly varying function, then the gradient is too noisy, and the saliency score provided by LEG is more meaningful. In Figure 1b, we show LEG for two different distributions. For the distribution with the larger variance, LEG evaluates the input's effect on the output for a larger neighborhood around $x_0$.

We note that the variance of $F$ has a large effect on LEG. As $F$ converges to a point mass at 0, if $f(x)$ is twice continuously differentiable in the neighborhood of $x_0$, then $\gamma \to \nabla_x f(x)$. On the other hand, if $F$ has high variance, then samples from $x_0 + F$ are substantially different from $x_0$ and LEG might no longer be useful for interpreting the model at $x_0$. This phenomenon can also described in terms of local vs global interpretation: for $F$ with a small variance, LEG provides a very local interpretation, i.e. a gradient that is valid in a small neighborhood around $x_0$, and as the variance of $F$ increases, LEG produces a more global interpretation, since a larger neighborhood around $x_0$ is considered in the calculation.

LEG has an analytical solution as the next lemma shows.

**Lemma 1.** *Let $Z$ be the random variable with a centered distribution $F$, i.e. $Z \sim F$ and $\mathbb{E}[Z] = 0_{p_1 \times p_2}$. Assume that covariance of $vec(Z)$ exists, and is positive-definite. Let $\Sigma = Cov(vec(Z))$, then*

$$\gamma(f, x_0, F) = vec^{-1} \left( \Sigma^{-1} \mathbb{E}_{z \sim F} \left[ (f(x_0 + z) - f(x_0)) \, vec(z) \right] \right). \tag{2}$$

Proof of the lemma is provided in the Appendix.

Lemma 1 shows that the LEG can be written as an affine transformation of a high dimensional integral where the integrand is $\int (f(x_0 + z) - f(x_0)) \, z F(z) dz$. This analysis also suggests an empirical estimate for the LEG, by replacing the expectation with the empirical mean. The empirical mean can be obtained by sampling $x$ from $F + x_0$, calculating $f(x)$, and then applying Lemma 1. More formally, let $x_1, \ldots, x_n$ be random samples from $F + x_0$, and let $y_1, \ldots, y_n$ be the function evaluations with $y_i = f(x_i)$. Further, let $\tilde{y}_i = f(x_i) - f(x_0)$ and $z_i = x_i - x_0$. Then, the empirical LEG estimate is given by

$$\hat{\gamma}(f, x_0, F) = \text{vec}^{-1}\left(\Sigma^{-1}\left[\frac{1}{n}\sum_{i=1}^{n}\text{vec}\left(\tilde{y}_i z_i\right)\right]\right). \tag{3}$$

As the function $f(x)$ is bounded and $F$ has a positive-definite covariance matrix, then it follows that as $n \to \infty$, $\hat{\gamma} \to \gamma$. However, classical linear model theory (Ravishanker & Dey, 2001) shows that rate of the convergence is very slow, on the order of $\frac{1}{\lambda_{\min}(\Sigma)}\sqrt{p_1 p_2/n}$, where $p_1$ and $p_2$ are the dimensions of $\mathcal{X}$. This severely limits the practicality of the empirical approach. In the next section we propose to use regularization in order to obtain faster convergence rates.

## 3 EFFICIENT ESTIMATION OF LEG

For interpretation of image classifiers, one expects that the saliency scores are located at a certain region, i.e. a contiguous body or a union of such bodies. This idea has lead to various procedures that estimate saliency scores by penalizing the local differences of the solution, often utilizing some form of the total variation (TV) penalty (Fong & Vedaldi, 2017). The approach is very sensible from a practical point of view: Firstly, it produces estimates that are easy to interpret as the important regions can be easily identified; secondly, penalization significantly shrinks the variance of the estimate and helps produce reliable solutions with less model evaluations.

In the light of the above, we propose to estimate the LEG coefficient with an anisotropic $L_1$ TV penalty.

**Definition 2** (LEG-TV). *For a hyperparameter, $L \geq 0$, the TV-penalized LEG estimate is given as $\tilde{\gamma} = vec^{-1}(g)$ where $g$ is the solution of the following linear program*

$$\min_g \|Dg\|_1$$
$$s.t. \left\|D^{+T}\left(\frac{1}{n}\sum_{i=1}^{n}vec\left(\tilde{y}_i z_i\right) - \Sigma g\right)\right\|_\infty \leq L, \tag{4}$$

*where $D \in \mathbb{R}^{(2p_1 p_2 - p_1 - p_2) \times (p_1 p_2)}$ is the differencing matrix with $D_{i,j} = 1, D_{i,k} = -1$ if the $j^{th}$ and the $k^{th}$ component of $g$ are connected on the two dimensional grid.*

Our method is based on the "high confidence set" approach which has been successful in numerous applications in high dimensional statistics (Candes & Tao, 2007; Cai et al., 2011; Fan, 2013). The set of $g$ that satisfy the constraint in the formulation is our high confidence set; if $L$ is chosen properly, this set contains the true LEG coefficient, $\gamma(f, x_0, F)$, with high probability[1]. This setup ensures that the distance between $\gamma$ and $\tilde{\gamma}$ is small. When combined with the TV penalty in the objective function, the procedure seeks to find a solution that both belongs to the confidence set and has sparse differences on the grid. Thus, the estimator is extremely effective at recovering $\gamma$ that have small total variation.

The proposed method enjoys low computational complexity. The problem in equation 4 is a linear program and can be solved in polynomial time, for instance by using a primal-dual interior-point method for which the time complexity is $O\left((p_1 p_2)^{3.5}\right)$ (Nocedal & Wright, 2006). However, in practice, solutions can be obtained much faster using simplex solvers. In our implementations, we use MOSEK, a commercial grade simplex solver by ApS (2019), and are able to obtain a solution in less than 3 seconds on a standard 8-core PC for a problem of size $p_1 = p_2 = 28$. Additionally, the alternative formulation (provided in the Appendix) can be solved using parametric simplex approaches which yield the whole solution path in $L$ (Vanderbei, 2014). The last point is often a necessity in deployment when $L$ needs to be tuned according to some criteria.

We note that the procedure does not require any knowledge about the underlying neural network and is completely model-agnostic. In fact, in applications where security or privacy could be a concern and returning multiple prediction values needs to be avoided, the term given by $\sum_{i=1}^{n}\text{vec}\left(\tilde{y}_i z_i\right)$ can be computed on the side and supplied alongside the prediction.

---
[1]See Lemma 2 in the appendix.

In Figure 2, we show the resulting estimates of the method with $n = 500$ model evaluations for a VGG-19 (Simonyan & Zisserman, 2014) network. For the distribution $F$, we use a multivariate Gaussian distribution with the proposed perturbation scheme in Section 4.2. We compute $\tilde{\gamma}$ separately for each channel, and then sum the absolute values of the different channels to obtain the final saliency score.

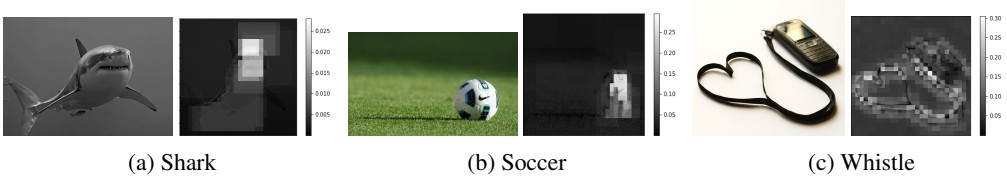

| (a) Shark | (b) Soccer | (c) Whistle |

Figure 2: LEG estimates for various images. Target classes are provided in the captions. LEG correctly detects the main object in all of the instances. Third image is labeled as a "cellphone" but the VGG-19 network misclassifies it as a "whistle". More results are provided in the Appendix.

## 4 THEORETICAL ANALYSIS AND IMPLEMENTATION

In this section, we analyze the procedure from a theoretical perspective and derive finite sample convergence rates of the proposed LEG-TV estimator. As we noted earlier, this analysis also gives us insight on the properties of the ideal perturbation distribution.

### 4.1 CONSISTENCY

We first present our condition, which has a major role in the convergence rate of our estimator. The condition is akin to the restricted eigenvalue condition (Bickel et al., 2009) with adjustments specific to our problem.

**Assumption 1.** *Let $D^+$ be the pseudo-inverse of the differencing matrix $D$, and denote the elements of singular value decomposition of $D$ as $U, \Theta, V$ where $D = U\Theta V^T$. Furthermore, denote the last $p_1 p_2 - p_1 - p_2$ columns of $U$ that correspond to zero singular values as $U_2$. For the covariance matrix $\Sigma$, and any set $S$ with size $s$, it holds that $\kappa > 0$, where*

$$\kappa = \inf_{\substack{\|\Delta_S\|_1 \geq \|\Delta_{S^C}\|_1 \\ U_2^T \Delta = 0}} \frac{\Delta^T D^{+T} \Sigma D^+ \Delta}{\|\Delta\|_2^2}. \tag{5}$$

The following theorem is our main result.

**Theorem 1.** *Let $\gamma^* = \gamma(f, x_0, F)$ and $\Sigma = Cov\,(vec(Z))$, where $Z \sim F$ and $\mathbb{E}[Z] = 0_{p_1 \times p_2}$. Let $\tilde{\gamma}$ be the LEG-TV estimate with $L = \sqrt{2\|D^+\|_1 \log\,(p_1 p_2/\epsilon)\,/n}$. If Assumption 1 holds for the covariance matrix $\Sigma$ with constant $\kappa$, then with probability $1 - \epsilon$,*

$$\left\|\gamma^* - \tilde{\gamma} - m 1_{p_1} 1_{p_2}^T\right\|_{Fr}^2 \leq \frac{1}{\kappa}\frac{C_p}{C_d}\sqrt{\frac{s \log p_1 p_2/\epsilon}{n}},$$

*where $m \in \mathbb{R}$ is a mean shift parameter, $s$ is the number of non-zero elements in $D\gamma^*$, $C_p = 4\sqrt{2\|D^+\|_1} \propto p_1^{1/4} p_2^{1/4}$ and $C_d$ is the minimal positive singular value of $D$.*

The proof is built on top of the "high confidence set" approach of Fan (2013). In the proof, we first establish that, for an appropriately chosen value of $L$, $\gamma^* = \gamma(f, x_0, F)$ satisfies the constraint in equation 4 with high probability. Then, we make use of TV sparsity of $\tilde{\gamma}$ and $\gamma^*$ to argue that the two quantities cannot be too far away from each other, since both are in the constraint set. The full proof is provided in the Appendix.

Our theorem has two major implications:

1. We can recover the true parameter as the number of model evaluations increase. That is, TV penalized LEG is a statistically consistent model interpretation scheme. Furthermore, our result states that, ignoring the log terms, one needs $n = O(s\,(p_1 p_2)^{1/2})$ many model evaluations to reliably recover $\gamma^*$.

2. Our bound depends on the constant $\kappa$, which further depends on the choice of $\Sigma$ for the perturbation scheme. It is possible to obtain faster rates of convergence with a carefully tuned choice of $\Sigma$. As a side note, since $\gamma^*$ also depends on $\Sigma$, the estimand changes when $\Sigma$ is adjusted. In other words, our result states that certain estimands require less samples.

We note that our procedure identifies the LEG coefficient up to a mean shift parameter, $m$, which is the average of the true LEG coefficient $\gamma$. In practice, the average can be consistently estimated (for instance, using the empirical version of LEG in equation 3), and the mean can be subtracted to yield consistent estimates for $\gamma$. However, in our numerical studies, we see that this mean shift is almost non-existent: LEG-TV yields solutions that has no mean differences with the LEG coefficient, which we define as the solution of the empirical version as $n \to \infty$.

## 4.2 PERTURBATION SCHEME

In our main result, we established that the convergence of our estimator depends on the quantity $\kappa$ which is related to the spectral properties of $\Sigma$. In this subsection we explore the ramifications of the assumption.

Our main result in Theorem 1 states that the rate of convergence to the true LEG coefficient is inversely proportional to the term $\kappa$. Thus, perturbation schemes for which the restricted eigenvalues are large, as defined in Definition 1, yield saliency maps that require less samples to estimate the LEG.

We note that most of the saliency estimation procedures that make use of perturbations take these perturbations to be independent, which results in a covariance matrix that is equal to the identity matrix, $\Sigma = \sigma^2 \mathbb{I}_{(p_1 p_2) \times (p_1 p_2)}$ for some $\sigma^2 > 0$. For LEG estimation without penalization, i.e. using equation 1, this choice is also optimal as the convergence rates under the normal setup depend on $1/\lambda_{\min}(\Sigma)$. However, when one seeks to find an estimate for which the solution is sparse in the TV norm, this choice is no longer ideal as demonstrated by our theorem.

In order to choose the covariance matrix of our perturbation scheme in a manner that maximizes the bound in equation 5, one also needs some prior information about the size of $S$, $s$. As that requires estimation of $s$, and a complex optimization procedure, we instead propose a heuristic: we choose $\Sigma$ so that its eigenvectors match $D^+\Delta$ for vectors $\Delta$ with unit-norm and $U_2^T\Delta = 0$. This choice fixes $p_1 p_2 - 1$ many of the eigenvectors of $\Sigma$. For the last eigenvector, we use the one vector as it is orthogonal to the rest of the eigenvectors. Our proposed perturbation scheme is as follows:

1. Compute the singular value decomposition of $D$, and let $D = U\Theta V^T$.

2. Let $\Sigma = \sigma^2\left(V\Theta^2 V^T + \frac{1}{p_1 p_2} 1_{p_1 p_2} 1_{p_1 p_2}^T\right)$ for some choice of $\sigma^2 > 0$.

As $D^+ = V\Theta^+ U^T$, with the proposed $\Sigma$, the numerator in equation 5 reduces to $\sigma^2 \Delta^T \Delta$ and hence $\kappa = \sigma^2$. Without any additional assumptions on $S$, this is the maximal value for $\kappa$.

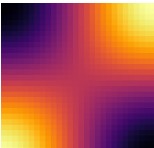 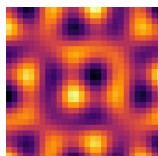 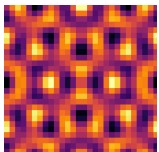 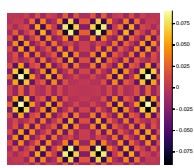

Figure 3: Selected eigenvectors of the proposed $\Sigma$. The eigenvectors, which contain the principal directions of the distribution, have maxima and minima in adjacent locations. Distributions drawn with these properties perform as object detectors as they can be used to detect existence (or non-existence) of significant pixels at these locations.

We plot some of the eigenvectors for our proposed $\Sigma$ with $p_1 = p_2 = 28$ in Figure 3. These eigenvectors are the principal directions of the perturbation distribution $F$, and the samples drawn from $F$ contain a combination of these directions. We see these samples will have sharp contrasts at certain locations. This result is very intuitive: The perturbation scheme is created for a specific problem where boundaries for objects are assumed to exist, and large jumps in the magnitude of the distribution help our method recover these boundaries efficiently.

We conclude this section with a demonstration of the perturbation scheme using Gaussian noise. In Figure 4, we plot a digit from the MNIST dataset (LeCun et al., 1998), along with instances obtained by independent perturbation and by our suggested distribution.

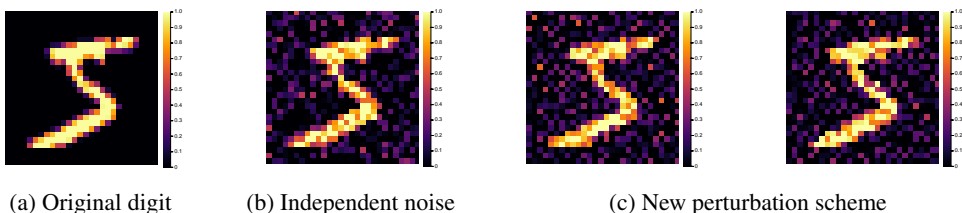

(a) Original digit      (b) Independent noise      (c) New perturbation scheme

Figure 4: Demonstration of the new perturbation scheme on an example from the MNIST dataset. Noise samples of the new scheme have a checkerboard pattern and their perturbations are uniformly distributed across the image.

### 4.3 IMPLEMENTATION DETAILS

LEG-TV procedure has two tuning parameters: (i) $F$, which determines the structure of the perturbation; and (ii) $L$, which controls the sparsity of the chosen interpretation.

Regarding $F$, we propose to use a multivariate Gaussian distribution as it is easy to sample from. For $\Sigma$, we propose a theoretically driven heuristic for determining the correlation structure of $\Sigma$ in Section 4.2. However, the choice of the magnitude of $\Sigma$, i.e. $\sigma^2$, is left to the user. If this quantity is chosen too low, then the added perturbations are small in magnitude, and the predictions of the neural network do not change, resulting in a LEG near zero. On the other hand, with a very large value of $\sigma^2$, the results have too much variance as some of the pixel values are set to the minimum or the maximum pixel intensity. In our implementations, we find that setting $\sigma^2$ to be between 0.05 and 0.30 results in reasonable solutions. We determine this range by computing perturbations of various sizes on numerous images using the VGG-19 classifier. The provided range is found to create perturbations large enough to change the prediction probabilities but small enough to avoid major changes in the image. Most of our presented results are given for $\sigma^2 = 0.10$.

For the choice of $L$, we propose two solutions: The first is the theoretically suggested quantity given in Theorem 1, although this often results in estimates that are too conservative. Our second method is a heuristic based on some of the quantities in the optimization problem and we use this for our demonstrations. We set $L = K_L L_{\max}$ where $K$ is a constant between 0 and 1 and $L_{\max}$ is the smallest value of $L$ for which the solution in equation 4 would result with $g = 0$; i.e. $L_{\max} = n^{-1} \| D^{+T} \left( \sum_{i=1}^{n} \text{vec}(\tilde{y}_i z_i) \right) \|$. We use $K_L = 0.05$ or $K_L = 0.10$ in our implementations. We note that is possible to obtain the solution for all $L$ by using a parametric simplex solver (Vanderbei, 2014), or by starting with a large initial $L$, then using the solution of the program as a warm-start for a smaller choice of $L$. Both approaches return the solution path for all $L$, and might be more desirable in practice than relying on heuristics.

## 5 EXAMPLES

In this section, we demonstrate the robustness and validity of our procedure by two numerical experiments. In Section 5.1, we perform sanity checks as laid out by Adebayo et al. (2018b), and show that the LEG-TV estimator fails to detect objects when the weights of the neural network are chosen randomly. In Section 5.2, we implement a sensitivity analysis in which we use various saliency methods to compute regions of importance, and then perturb these regions in order to see their effect on the prediction. For the deep learner, we use VGG-19 (Simonyan & Zisserman, 2014). For

computational efficiency, we compute saliency maps on a 28 by 28 grid (i.e. $\tilde{\gamma} \in \mathbb{R}^{28 \times 28}$) although the standard input for VGG-19 is 224 by 224. The perturbations on the image are scaled up by 8 via upsampling in order for the dimensions to match.

## 5.1 SANITY CHECKS

In Adebayo et al. (2018b), the validity of saliency estimation procedures are tested by varying the weights of the neural network. In a technique named, "cascading randomization", authors propose to replace the fitted weights of a CNN layer by layer, and compute the saliency scores with each change. As a deep learner with randomly chosen weights should have no prediction power, one expects to see the same effect in the resulting saliency scores: namely, as more of the weights are perturbed, the explanation offered by interpretability methods should become more and more meaningless. Surprisingly, Adebayo et al. (2018b) show that most commonly adopted interpretation procedures provide some saliency even after full randomization, and conclude that these methods act as edge detectors.

Our procedure treats the classifier as a black-box and the explanations offered by LEG-TV are based solely on the predictions made by the neural network. During the sanity check, when the weights of the neural network are randomly perturbed, the predictions change significantly and no longer depend on the input. Thus, we expect the local linear approximations of the underlying function to be flat, which would result in saliency scores of zero for all of the pixels. Finally, small artifacts that might arise in this process, such as positive or negative saliency scores with no spatial structure, should be smoothed over due to the TV penalty, further robustifying our procedure.

In order to verify our intuition, we perform cascading randomization on the weights of a VGG-19 network. For all of the images in our analysis, we find that the LEG-TV estimate, $\tilde{\gamma}$, is reduced to zero after randomization of either the top (i.e. logits) or the second top layer (i.e. second fully connected layer). The results of our experiment for two images are given in Figure 5. It is seen that after the weights are perturbed, the LEG-TV method fails to detect any signal that could be used for interpretation. In fact, due to penalization, the estimate is set to zero. These results show that the interpretation given by our proposed method is reliable and is dependent on the classifier.

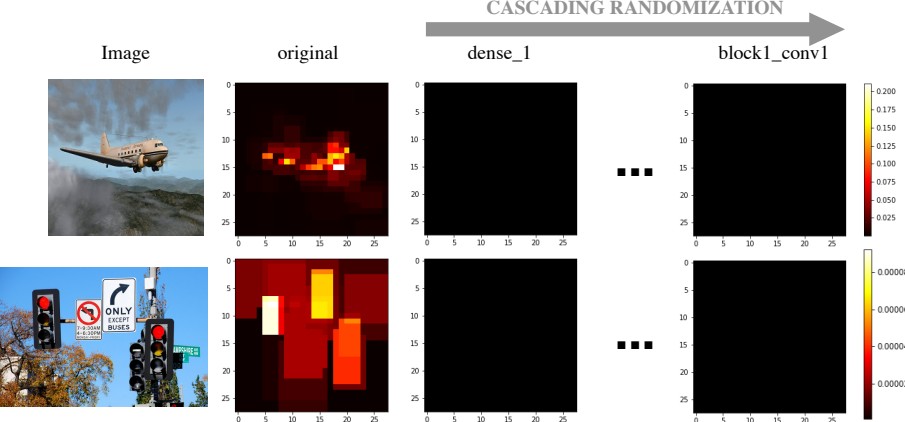

Figure 5: Results of the sanity check with cascading randomization. The network weights are replaced by random numbers in a cascading order, starting from the last layer. LEG is equal to zero for all pixel values immediately after the first randomization.

## 5.2 SENSITIVITY ANALYSIS

For our second validity test, we use various interpretation models to compute regions of high importance. We then mask these regions by decreasing the value of the pixels to zero which is equivalent to painting them black. We compute and assess the difference of the predictions for the target class with each perturbation.

We compare our method against four alternatives: GradCAM (Selvaraju et al., 2017), LIME (Ribeiro et al., 2016), SHAP (Lundberg & Lee, 2017) and C-Shapley (Chen et al., 2019). The last three methods are chosen as they are model-agnostic, like LEG, and do not make use of the architecture of the neural network. GradCAM is chosen due to its popularity.

The saliency maps using C-Shapley and LEG-TV are computed for a 28 by 28 grid. In order to make the comparison between the methods more fair, we downsize the saliency maps resulting from GradCAM, LIME and SHAP to the same size. Interestingly, we find that this step improves the performance of these estimators; that is, the perturbations identified using the low resolution saliency maps result in faster drops in the predicted score. For LEG-TV, LIME and SHAP, the saliency scores are computed using 3000 model evaluations, where as C-Shapley requires 3136 ($28\times28\times4$) evaluations. For LEG-TV, we provide two solutions, a sparse solution which corresponds to a larger choice of the penalty parameter $L$ and a noisy solution which is obtained with a smaller choice of $L$, denoted by LEG and LEG0, respectively. We present the results for 500 images that are randomly chosen from a subsample of the ImageNet dataset (Deng et al., 2009)[2]. The average of the log odds ratios across the 500 images are provided in Figure 6. We see that as the size of the perturbation increases, the predictions for the target class drop for all of the methods. The slope is sharpest for SHAP and LEG0, suggesting that these two methods identify pixels that are crucial for the predictions.

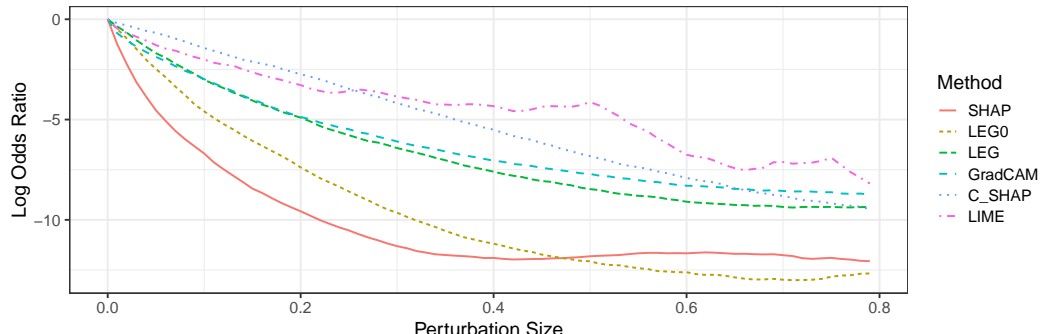

Figure 6: Results of sensitivity analysis. Log of the predicted probability for the target class is plotted versus the size of the perturbation. The locations for the perturbations are determined by the saliency procedures. Predictions should decrease at a fast rate for interpretability methods that can reliably identify regions of importance. In that regard, SHAP and LEG0 appear to be the most accurate in determining the critical pixels, followed by LEG, GradCAM, C-Shapley and LIME.

In Figure 7, we plot the top 10% most salient pixels according to different procedures for three images in the dataset. The pixels chosen by SHAP appear to correspond to specific a convolution pattern and the chosen region is not contiguous. On the other hand, pixels identified by LEG-TV are visually meaningful to the human eye and contain pixels that are more likely to be relevant for the prediction. LEG-TV selects different parts of the crane in the first image, and the face of the Pekinese dog in the second. In the last image, where a soap dispenser is misclassified as a soda bottle, LEG-TV relates the classification to the label and the barcode of the bottle – parts that are often seen on soda bottles. For the same image, LEG-TV also selects the fixtures in the background, which could have been mistaken by the classifier as the cap of the soda bottle.

## 6  DISCUSSION

We have proposed a statistical framework for saliency estimation that relies on local linear approximations. Utilizing the new framework, we have built a computationally efficient saliency estimator that has theoretical guarantees. Using our theoretical analysis, we have identified how the sample complexity of the estimator can be improved by altering the model evaluation scheme. Finally, we

---

[2]The dataset is provided by fastai (Howard et al., 2019) and can be found at http://files.fast.ai/data/imagenet-sample-train.tar.gz

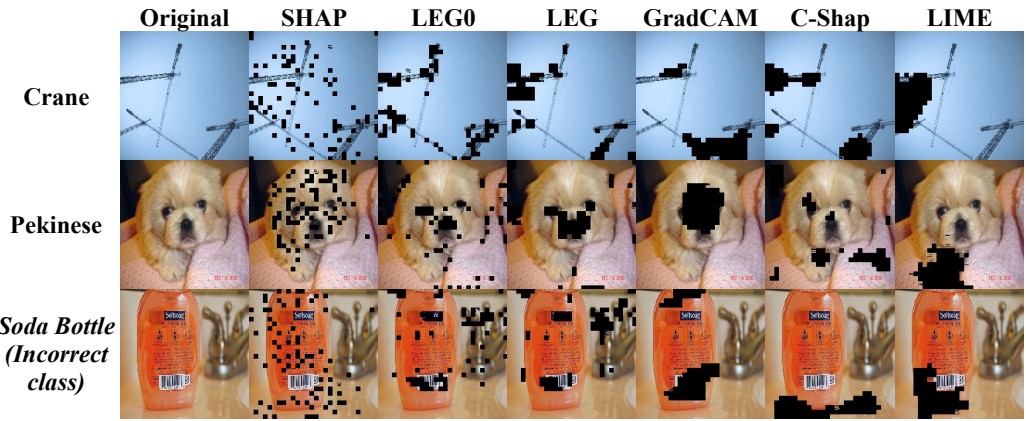

Figure 7: Masked regions at 10% perturbation by various saliency procedures.

have shown through empirical studies that (i) unlike most of its competitors, our method passes the recently proposed sanity checks for saliency estimation; and (ii) pixels identified through our approach are highly relevant for the predictions, and our method often chooses regions with higher saliency compared to regions suggested by its alternatives.

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

# A  APPENDIX

## A.1  ALTERNATIVE FORMULATION

Our linear program can also be recast by a change of variables and setting $\alpha = Dg$. In this case, the elements of $\alpha$ correspond to differences between adjoint pixels. This program can be written as:

$$\min \|\alpha\|_1$$

$$\text{s.t.} \left\| D^+ \left( \frac{1}{n} \sum_{i=1}^n \tilde{f}(\tilde{x}_i) \tilde{x}_i - \Sigma D^+ \alpha \right) \right\|_\infty \leq L,$$

$$U_2^T \alpha = 0,$$

where $D^+$ is the pseudo-inverse of $D$ and $U_2$ is related to the left singular vectors of $D$. More precisely, letting $D = U\Theta V^T$ denote the singular value decomposition of $D$, $U_2$ is the submatrix that corresponds to the columns of $U$ for which $\Theta_j$ is zero. The linearity constraint ensures that the differences between the adjoint pixels is proper. Derivation of the alternative formulation follows from Theorem 1 in Gaines et al. (2018) and is omitted.

This formulation can be expressed in the standard augmented form, i.e. $\min_{Ax=b, x \geq 0} c^T x$, by writing $x = [\alpha_+, \alpha_-, s_+, s_-]^T$,

$$A = \begin{bmatrix} U_2 & -U_2 & 0 & 0 \\ -D^+\Sigma D^+ & D^+\Sigma D^+ & \mathbb{I}_{m\times m} & 0 \\ -D^+\Sigma D^+ & D^+\Sigma D^+ & 0 & -\mathbb{I}_{m\times m} \end{bmatrix}, \qquad b = \begin{bmatrix} 0 \\ L1_m - D^+y \\ -L1_m - D^+y \end{bmatrix}, \qquad c = \begin{bmatrix} 1_m \\ 1_m \\ 0 \\ 0 \end{bmatrix},$$

where $y = \frac{1}{n} \sum_{i=1}^n \tilde{f}(\tilde{x}_i) \tilde{x}_i$ and $m = 2p_1p_2 - p_1 - p_2$. The $\gamma$ coefficient in the original formulation can be obtained by setting $\gamma = D^+ (\alpha_+ - \alpha_-)$.

## A.2  PROOF OF THEOREM 1

Our proof depends on the following lemma.

**Lemma 2.** *For $L \geq \sqrt{2\|D^+\|_1 \log(p_1p_2/\epsilon)/n}$, $\gamma^*$ is in the feasibility set with probability $1 - \epsilon$, that is*

$$\left\| D^+ \left( \frac{1}{n} \sum_{i=1}^n \tilde{f}(\tilde{x}_i) \tilde{x}_i \right) - D^+\Sigma\gamma^* \right\|_\infty \leq L.$$

*Proof.* For ease of notation, let $G = D^+\mathbb{E}\left[ \frac{1}{n} \sum_{i=1}^n \tilde{f}(\tilde{x}_i) \tilde{x}_i \right]$, and note that $G = D^+\Sigma\gamma^*$. Furthermore, let $z_i = \tilde{f}(\tilde{x}_i) D^+\tilde{x}_i$. We also assume that the images have been rescaled so that the maximum value of $\tilde{x}_i$ is 1 (without rescaling, the maximum would be given as the largest intensity, i.e. 255). Since, the function values are also in the range given by [-2,2], we can bound $|z_{i,j}|$, that is

$$|z_{i,j}| = \left| \tilde{f}(\tilde{x}_i) D_j^+ \tilde{x}_i \right| \leq 2 \left\| D_j^+ \right\|_1 \max_i |x_{i,j}| \leq 2 \left\| D_j^+ \right\|_1.$$

The proof follows by applying the McDiarmid's inequality (Vershynin, 2018) for each row of the difference and then taking the supremum over the terms. By application of McDiarmid's inequality, we have that

$$\mathbb{P}\left(\left|\frac{1}{n}\sum_i z_{ij} - G_j\right| \geq L\right) \leq 2e^{\frac{-L^2 n}{2\|D^+\|_1}}.$$

Let $L = \sqrt{2\|D^+\|_1 \log\left(p_1 p_2/2\epsilon\right)/n}$. Then, taking a union bound over all variables, we have

$$\mathbb{P}\left(\max_j \left|\frac{1}{n}\sum_i z_{ij} - G_j\right| \geq L\right) \leq \sum_{j=1}^p e^{\frac{-L^2 n}{2\|D^+\|_1}} = \epsilon.$$

Now note that that the feasibility set for any $L' \geq L$ contains that of $L$ and thus $\gamma^*$ is automatically included. $\qquad\square$

We now present the proof of the theorem. Note that the technique is based on the Confidence Set approach by Fan (2013). In the proof, we use $\gamma$ to refer to $\text{vec}(\gamma)$ for ease of presentation.

*Proof.* First, let the high probability set for which Lemma 2 holds by $A$. All of the following statements hold true for $A$. We let $\Delta = D\left(\hat{\gamma} - \gamma^*\right)$. We know that $\|D\hat{\gamma}\|_1 \leq \|D\gamma^*\|_1$ since both are in the feasibility set, as stated in Lemma 2. Let $\alpha^* = D\gamma^*$, $\hat{\alpha} = D\hat{\gamma}$ and define $S = \{j : \alpha_j^* \neq 0\}$, and the complement of $S$ as $S^C$. By assumption of the Theorem, we have that the cardinality of $S$ is $s$, i.e. $|S| = s$. Now let $\Delta_S$ as the elements of $\Delta$ in $S$. Then, using the above statement, one can show that $\|\Delta_S\|_1 \geq \|\Delta_{S^C}\|_1$. Note,

$$\begin{aligned}
\|\hat{\alpha}\|_1 &= \|\alpha^* + \Delta\|_1 \\
&= \|\alpha^* + \Delta_S\|_1 + \|\Delta_{S^C}\|_1 \\
&\geq \|\alpha^*\|_1 - \|\Delta_S\|_1 + \|\Delta_{S^C}\|_1 \\
&\geq \|\hat{\alpha}\|_1 - \|\Delta_S\|_1 + \|\Delta_{S^C}\|_1,
\end{aligned}$$

and $\|\Delta_S\|_1 \geq \|\Delta_{S^C}\|_1$ follows immediately. Furthermore

$$\left\|\hat{\Delta}\right\|_2 \geq \left\|\hat{\Delta}_S\right\|_2 \geq \left\|\hat{\Delta}_S\right\|_1 / \sqrt{s} \geq \frac{\left\|\hat{\Delta}\right\|_1}{2\sqrt{s}},$$

where the last line uses the previous result.

Additionally, note that

$$\begin{aligned}
\Delta^T D^+ \Sigma D^+ \Delta &\leq \|\Delta\|_1 \|D^+ \Sigma D^+ \Delta\|_\infty \\
&\leq 2L\|\Delta\|_1,
\end{aligned}$$

where the first inequality follows by Holder's inequality and the second follows from Lemma 2 and the fact that both $\hat{\gamma}$ and $\gamma^*$ are in the feasibility set for $L = \sqrt{2\|D^+\|_1 \log\left(p_1 p_2/\epsilon\right)/n}$. We further bound the right hand side of the inequality by using the previous result, which gives

$$\Delta^T D^+ \Sigma D^+ \Delta \leq 4L\sqrt{s}\|\Delta\|_2.$$

Next, we bound $\|\Delta\|_2$ by combining the previous results. Now, by assumption of the Theorem, we have that

$$\begin{aligned}
a\|\Delta\|_2^2 &\leq \Delta^T D^{+T} \Sigma D^+ \Delta \\
&\leq 4L\sqrt{s}\|\Delta\|_2.
\end{aligned}$$

Dividing both sides by $\|\Delta\|_2$, we obtain that

$$\|D\hat{\gamma} - D\gamma^*\|_2 \leq \frac{C_p}{a}\sqrt{\frac{s \log p_1 p_2/\epsilon}{n}}.$$

Finally, we note that

$$\|D(\hat{\gamma} - \gamma^*)\|_2^2 = \|D(m1 + \hat{\gamma} - \gamma^*)\|_2^2$$

$$\geq C_D \|m1 + \hat{\gamma} - \gamma^*\|_2^2 + \frac{1}{p_1 p_2} \left( p_1 p_2 m + \sum_j \tilde{\gamma}_j - \sum_j \gamma_j^* \right)^2,$$

where $D$ is the smallest singular value of $D$ that is positive. This follows from the fact that $D$ has only one zero right singular value, whose eigenvector is given by a vector of ones multiplied by $1/\sqrt{p_1 p_2}$. Letting $m = (p_1 p_2)^{-1} \left( \sum_j \gamma_j^* - \sum_j \tilde{\gamma}_j \right)$ concludes the proof.

$\square$

## A.3 PROOF OF LEMMA 1

*Proof.* Let

$$h(g) = \mathbb{E}_{x \sim F + x_0} \left[ \left( f(x) - f(x_0) - \text{vec}(g)^T \text{vec}(x_0 - x) \right)^2 \right].$$

Note that $h(g)$ is quadratic and convex in $g$. Taking the derivative with respect to $\text{vec}(g)$, and setting it to zero we obtain

$$\mathbb{E}_{x \sim F + x_0} \left[ -2 \text{vec}(x_0 - x) \left( f(x) - f(x_0) - \text{vec}(x_0 - x)^T \text{vec}(g^*) \right) \right] = 0,$$

where $g^*$ is the minimizer. After reorganizing the terms and setting $z = x - x_0$, we get

$$\mathbb{E}_{z \sim F} \left[ \text{vec}(z) \left( f(x_0 + z) - f(x_0) \right) \right] = \mathbb{E}_{z \sim F} \left[ \text{vec}(z) \text{vec}(z)^T \text{vec}(g^*) \right] = \Sigma \text{vec}(g^*),$$

where we use that $\Sigma = \text{Cov}(\text{vec}(z))$ in the last equation. The result follows trivially. $\square$

## A.4 EQUIVALENCY OF LEG-TV WITH EMPIRICAL LEG IF $L = 0$

**Lemma 3.** *For the LEG-TV estimate with $L = 0$, if the one vector is an eigenvector of $\Sigma$, i.e. $\Sigma 1_{p_1 p_2} = \lambda 1_{p_1 p_2}$, then the solution is equal to the empirical LEG estimate up to a location shift. That is, $\tilde{\gamma} = \hat{\gamma} + a 1_{p_1 p_2}$, for some $a \in \mathbb{R}$.*

Before the proof, we note that the eigenvector condition on $\Sigma$ can satisfied either with independent noise or our suggested scheme in Section 4.2.

*Proof.* Note that, if $L = 0$, then we have that

$$D^{+T} \left( \frac{1}{n} \sum_{i=1}^{n} \text{vec}(\tilde{y}_i z_i) \right) = D^{+T} \Sigma g.$$

As the only right singular vector of $D^{+T}$ with zero singular value is the one vector, the above statement is true iff

$$\frac{1}{n} \sum_{i=1}^{n} \text{vec}(\tilde{y}_i z_i) = \Sigma g + c 1_{p_1 p_2},$$

for some $c \in \mathbb{R}$. Solving for $g$, we obtain,

$$g = \Sigma^{-1} \left( \frac{1}{n} \sum_{i=1}^{n} \text{vec}(\tilde{y}_i z_i) - c 1_{p_1 p_2} \right) = \Sigma^{-1} \frac{1}{n} \sum_{i=1}^{n} \text{vec}(\tilde{y}_i z_i) - c \Sigma^{-1} 1_{p_1 p_2} = \hat{\gamma} - \frac{c}{\lambda} 1_{p_1 p_2},$$

where we use the fact that the one vector is an eigenvector of $\Sigma^{-1}$ with eigenvalue $\lambda^{-1}$. Setting $a = -\frac{c}{\lambda}$ concludes the proof.

$\square$

## A.5   EXAMPLES ON MNIST

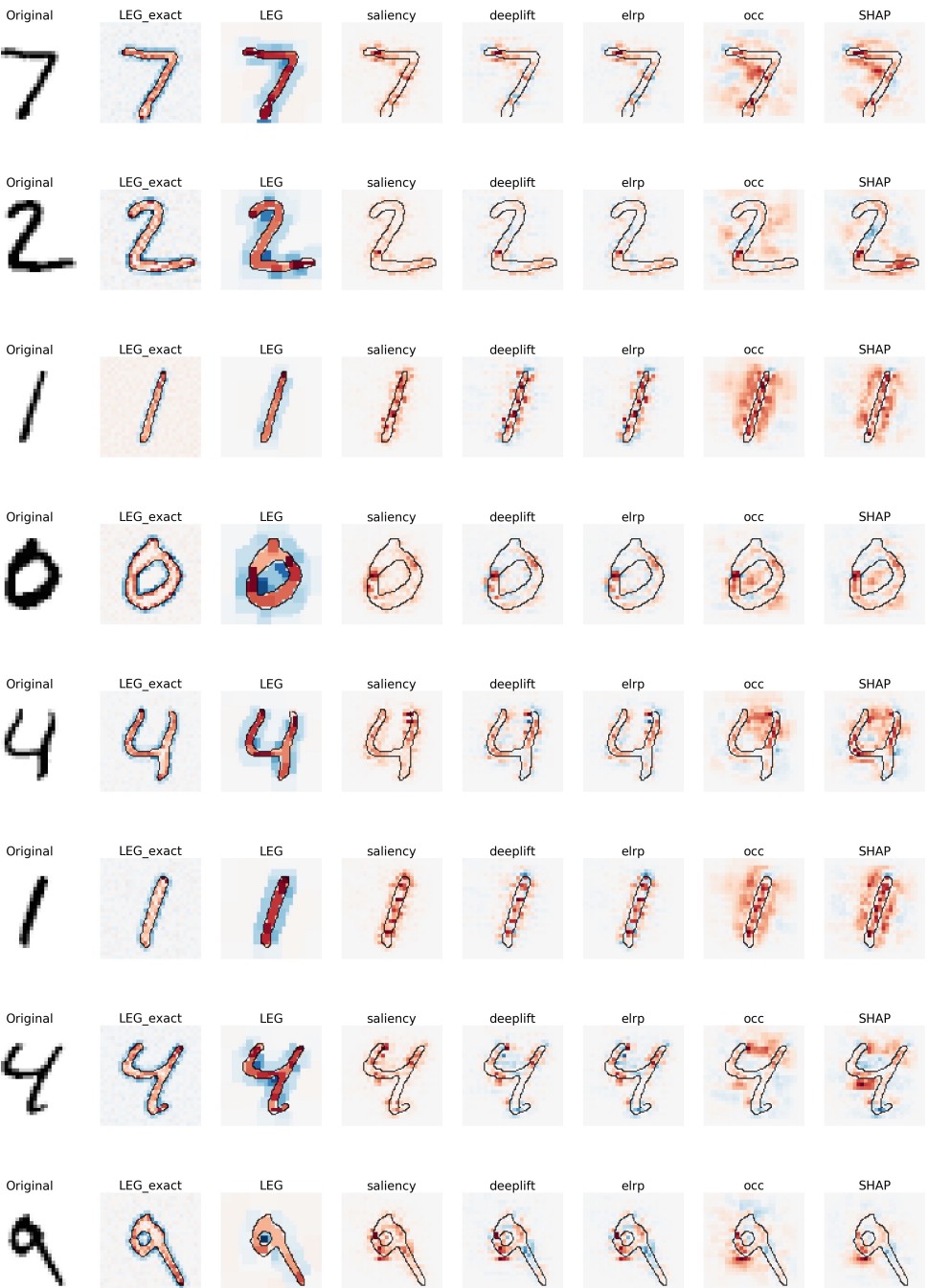

Figure 8: Saliency estimates from various procedures on the MNIST dataset for a LeNet (LeCun et al., 1998). The listed procedures are the empirical version of LEG [LEG-Exact], LEG, Direct Saliency, DeepLIFT (Shrikumar et al., 2017), ELRP (Bach et al., 2015), Occlusion Maps and SHAP (Lundberg & Lee, 2017)

## A.6 EXAMPLES ON IMAGENET

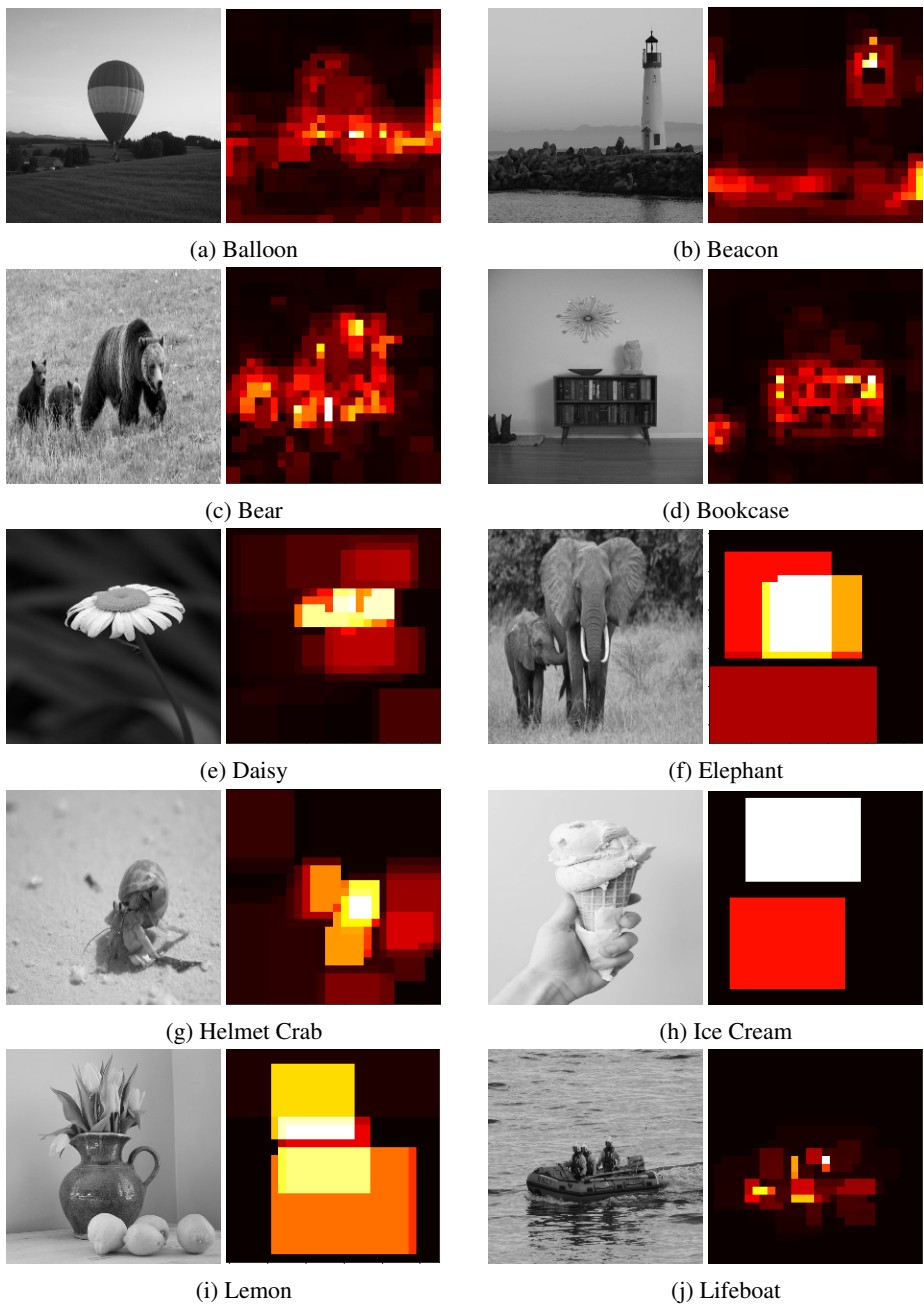

(a) Balloon

(b) Beacon

(c) Bear

(d) Bookcase

(e) Daisy

(f) Elephant

(g) Helmet Crab

(h) Ice Cream

(i) Lemon

(j) Lifeboat

Figure 9: LEG estimates for various images from the ImageNet dataset. Target classes are provided in the captions.

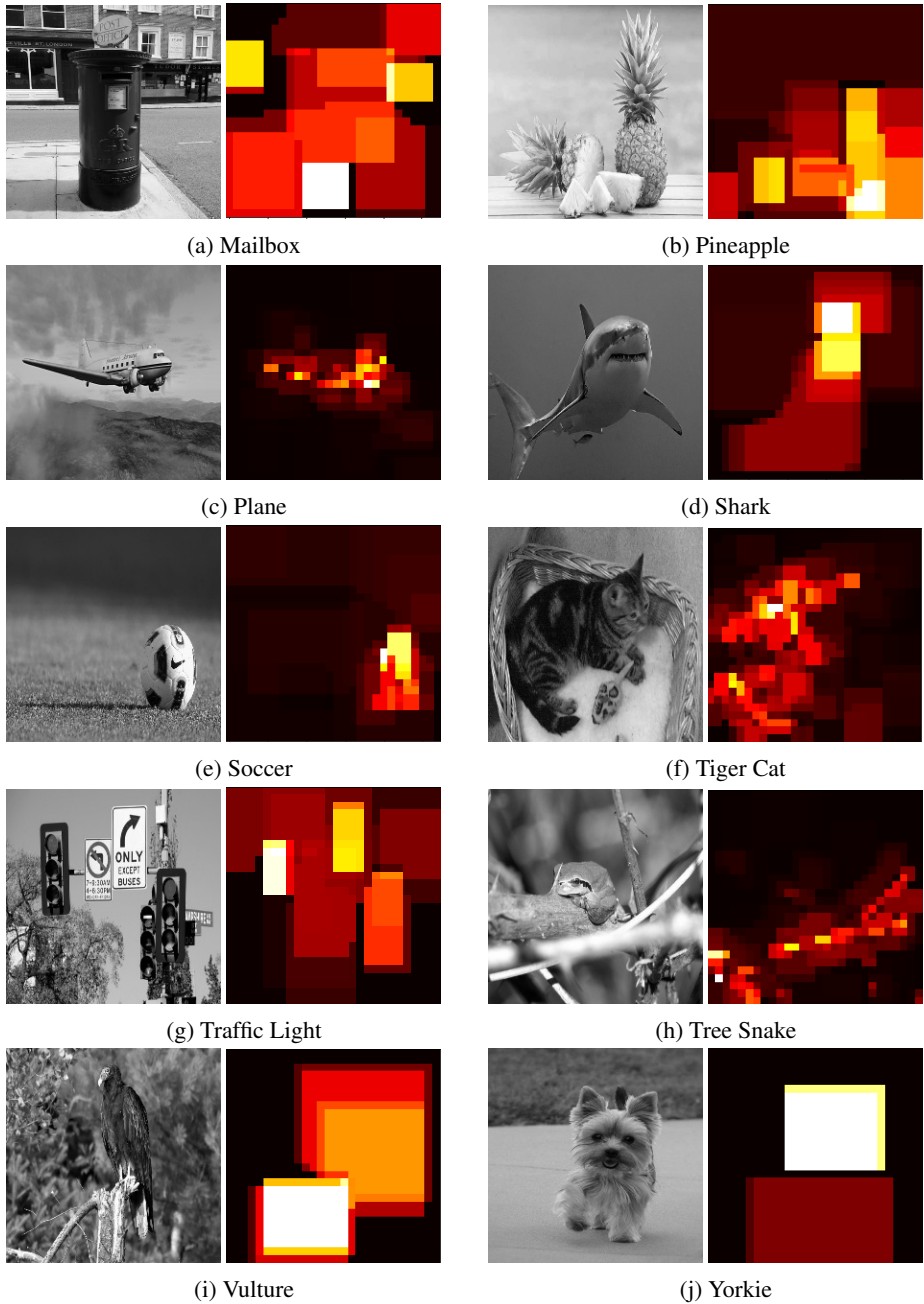

Figure 10: LEG estimates for various images from the ImageNet dataset. Target classes are provided in the captions.

