# OpenReview forum: "Statistically Consistent Saliency Estimation"
_ICLR.cc/2020/Conference — Reject_

### Official Review · AnonReviewer2 · 2019-10-22
**Official Blind Review #2**

**Rating:** 3

**Review:**

Authors proposes an interesting statistical method to detect saliency in images. Authors provides a specific estimator that is fast to compute and characterize its performance w.r.t. parameters.

My main concern is the experiment section. "For computational efficiency, we compute saliency maps on a 28 by 28 grid (i.e. γ˜ ∈ R 28×28) although the standard input for VGG-19 is 224 by 224. T". Shouldn't we do the same thing for all baselines? The seemingly good sailency results might stem from this artifact.

**Experience Assessment:**

I do not know much about this area.

**Review Assessment: Checking Correctness Of Derivations And Theory:**

I assessed the sensibility of the derivations and theory.

**Review Assessment: Checking Correctness Of Experiments:**

I carefully checked the experiments.

**Review Assessment: Thoroughness In Paper Reading:**

I read the paper at least twice and used my best judgement in assessing the paper.

---

> ### Author Response · Authors · 2019-11-09
> **Response to Reviewer 2**
>
> We thank the reviewer for the comment!
>
> Following your suggestion, we downsized the saliency maps from the other procedures and repeated the sensitivity analysis. The new results can be found in the updated paper. We found that this step significantly improves the performance of our competitors. We made the following addition to the text:
>
> "In order to make the comparison between the methods more fair, we downsize the saliency maps resulting from GradCAM, LIME and SHAP to a 28 by 28 grid. Interestingly, we find that this step improves the performance of these estimators."

---

### Official Review · AnonReviewer3 · 2019-10-23
**Official Blind Review #3**

**Rating:** 6

**Review:**

Author propose a statistical framework and a theoretically consistent procedure for saliency estimation that is close to the empirical solution and has sparse differences on the grid. I think the idea of the paper is quite interesting and the results are significant.

In particular, having an upper bound on the number of model evaluations to recover the region of
importance with high probability is significant. Moreover, they have proposed a new perturbation scheme for estimation of gradients that works better than random perturbation schemes.

I have some questions listed below:
- Can the proposed approach be integrated with different types of Saliency map methods?

- Can you explain the differences of the proposed approach and the group feature formulation of https://arxiv.org/abs/1902.00407 ?

- Is there a quantitative way to assess the performance of the proposed approach?

**Experience Assessment:**

I have published in this field for several years.

**Review Assessment: Checking Correctness Of Derivations And Theory:**

I assessed the sensibility of the derivations and theory.

**Review Assessment: Checking Correctness Of Experiments:**

I assessed the sensibility of the experiments.

**Review Assessment: Thoroughness In Paper Reading:**

I made a quick assessment of this paper.

---

> ### Author Response · Authors · 2019-11-09
> **Response to Reviewer 3**
>
> We thank the reviewer for the constructive comments and the questions! Please see our answers below.
>
> > Can the proposed approach be integrated with different types of Saliency map methods?
>
> - Certainly! One such integration is to use the proposed perturbation scheme in other perturbation based saliency methods, such as SmoothGrad, VarGrad or SmoothCASO (from the paper that you cited). An another possible avenue is to combine the constrained penalization approach with other saliency methods that utilize some sort of loss function. For instance, one can build a saliency estimation procedure which seeks to find a solution where the infinity-norm of the gradient of the loss is bounded. Our proof technique can be used to obtain theoretical consistency for such procedures.
>
> > Can you explain the differences of the proposed approach and the group feature formulation of https://arxiv.org/abs/1902.00407 ?
>
> - The group feature formulation approach of Singla et al (2019) seeks find pixels (or more precisely, perturbations) whose alteration will cause misclassification, and hence are more likely to be salient. There are two major differences between our work and theirs. Firstly, their method requires knowledge of the underlying neural network, as it uses the gradient and the hessian of the loss function at the input. Our procedure treats the network as a black-box, and can be utilized without any information about the network architecture or weights. Secondly, our approach for finding the salient points are different: our method seeks to approximate the predictor at a neighborhood around a specific input, and uses that information to identify the salient pixels; where as Singla et al provide local interpretations by finding ideal perturbations for a specific input. They also offer smoothed versions of their procedures, which help them obtain more interpretable solutions, but they don't have theoretical consistency results that would help them establish bounds on sample complexity. Due to its relevance, we also added this work, along with the work of Fong and Vedaldi (2017), to the literature review in our introduction.
>
> > Is there a quantitative way to assess the performance of the proposed approach?
>
> - Unfortunately, it is hard to find a single metric that can properly quantify how good an interpretation is. Most papers in the literature provide visual comparisons, and those are impossible to judge objectively. Furthermore, as different saliency methods estimate completely different quantities, there are no baseline problems with a specific ground-truth on which methods can be evaluated and compared against each other. We have seen two techniques in the literature that provide fully quantitative comparisons: (i) Sensitivity analyses based on perturbations; and (ii) Sanity checks. We include both of these in our numerical experiments.

---

### Official Review · AnonReviewer1 · 2019-10-23
**Official Blind Review #1**

**Rating:** 6

**Review:**

This work proposes a statistical framework for saliency estimation for black box
computer vision models, based on solving a convex program in (4). It also gives theoretical analysis on its consistency in Theorem 1, and run a few simulations to show the empirical performance of the proposed method.

The method proposed seems to be novel and reasonable. As a result, I tend to accept this paper. However, I would like to remark that solving (4) might be empirically difficult (depending on the size of the problem) even though it is convex and can be solved in polynomial time theoretically. I wonder if the authors could clarify the setup of their experiments (instead of writing "The problem in equation 4 can be
solved by any linear programming software, for which many open-source implementations exist"), and if the author could remark on the empirical running time.

Also, I am not sure if "Note that if L = 0, then the TV-penalization has no effect and the solution of the above procedure reduces to the empirical estimate," as the objective function is in L1 norm.

**Experience Assessment:**

I do not know much about this area.

**Review Assessment: Checking Correctness Of Derivations And Theory:**

I assessed the sensibility of the derivations and theory.

**Review Assessment: Checking Correctness Of Experiments:**

I did not assess the experiments.

**Review Assessment: Thoroughness In Paper Reading:**

I read the paper at least twice and used my best judgement in assessing the paper.

---

> ### Author Response · Authors · 2019-11-09
> **Response to Reviewer 1**
>
> We thank the reviewer for the constructive comments and the suggestions! Please see our comments below.
>
> > "However, I would like to remark that solving (4) might be empirically difficult (depending on the size of the problem) even though it is convex and can be solved in polynomial time theoretically. I wonder if the authors could clarify the setup of their experiments (instead of writing "The problem in equation 4 can be solved by any linear programming software, for which many open-source implementations exist"), and if the author could remark on the empirical running time."
>
> - In our numerical experiments we used the MOSEK solver. We edited the paper to clarify this point and included the empirical running time, which is less than 3 seconds on an 8 core computer. We also included the time complexity for interior point methods: one can obtain an $\epsilon$-accurate solution in $O((p_1p_2)^{3.5}\log(1/\epsilon))$ time. We note that it is possible to obtain much faster rates by utilizing the sparseness of the constraint matrix (Yen et al, 2015) or by using recently proposed stochastic central path methods (Cohen et al, 2018). We leave those as future directions of research.
>
> > "Also, I am not sure if "Note that if L = 0, then the TV-penalization has no effect and the solution of the above procedure reduces to the empirical estimate," as the objective function is in L1 norm."
>
> - This is correct. However, when $L=0$, the constraint reduces to an equality. As this equality is under determined, TV-penalization still plays an effect. The correct statement should be "the coefficient is equal to the empirical estimate up to a location shift". We found that this addition would make the paper harder to read, and we removed that comment. The proof for the above statement can be found in the appendix as an additional lemma (Lemma 3).
>
> References:
> - Yen, Ian En-Hsu, et al. "Sparse linear programming via primal and dual augmented coordinate descent." Advances in Neural Information Processing Systems. 2015.
> - Cohen, Michael B., Yin Tat Lee, and Zhao Song. "Solving linear programs in the current matrix multiplication time." Proceedings of the 51st Annual ACM SIGACT Symposium on Theory of Computing. ACM, 2019.

---

### Official Review · AnonReviewer4 · 2019-10-29
**Official Blind Review #4**

**Rating:** 8

**Review:**

The authors propose a nice framework for interpreting differentiable models without access to model details. The framework also comes with two empirical estimates, with solid theoretical back-up.

Definition 1 in Section 2 provides a definition for the method to study (LEG). LEG is a nice generalization of multiple existing approaches. There also seems to be a lot more to investigate for future work based on this framework.

After the authors proposed Definition 2 (seems to be a bit too straightforward, to be discussed later), Theorem 1 is proposed to characterize its convergence. It also provides guidelines for selecting the covariance matrix $\Sigma$, as discussed in Section 4.

Overall, the paper proposes a nice framework for model interpretation. It is well backed up by theory. It should clearly be accepted, although the paper is relatively weak in experiments. Below I will discuss some weaknesses and potential improvements.

Weakness:
1. Theorem 1 provides a nice characterization for the property of the proposed LEG-TV estimate. But there are two weaknesses.

First, the authors should note that the proposed definition 2 is still a bit too straightforward without intuitive explanations. For example, what is the general form of 2D Fused Lasso? How is it applied to approximate Definition 1 / Equation (2) to get the expression in Definition 2? More details may be explained to help readers understand.

Second, it should be carefully discussed that the proof of Theorem 1 depends on existing work if the connection is close. For example, the authors may add in the main content "the proof of Theorem 1 is built on top of ..." or statements like that, probably with a short sketch / intuition on the entire proof if space permits.

2. Can the authors be more specific on the time complexity and sample complexity of the proposed algorithms?

3. It seems the experimental section of the paper is not satisfactory. Almost all results are single-image analysis, instead of systematic empirical analysis on an image data set. Without such analysis, it is hard to see the advantage of the proposed method over other comparing saliency maps and model-agnostic methods. For example, is the proposed method more sample-efficient than LIME or SHAP, or other more efficient procedures such as L(C)-Shapley? Is the proposed method really selecting meaningful segments for the model? (It may be tested by evaluating the log-odds-ratio after the top selected features are masked.) It is observed that LEG is able to select connected regions (Figure 6). The same phenomenon has been observed for C-Shapley. It may be helpful if the connection is discussed (such as the connection between sampling procedure of C-Shapley and the procedure imposed by LEG-TV).

The reviewer is not conditioning the "accept" decision on adding any of the suggested improvements on experiments given the limited rebuttal period and limited space, although it may benefit the paper of some of them can be addressed.
-----------------------------------------------------------------------------------------------------
--------------------------------------Post Author Response--------------------------------
-----------------------------------------------------------------------------------------------------

The authors have addressed most of my concerns. Theorem 1 and the sketch of the proof have been discussed. Also, complexity has been discussed (the definition of $s$ is embedded in the theorem and may be made clearer). Last but not least, authors have carried out experiments in a larger scale to get more stable results.

      The performance, in terms of log-odds-ratio, may not be as good as some of the comparing methods. However, the paper provides a creative framework for incorporating structure into feature attribution scores in model interpretation. So I will keep my score.

      A typo: The first paragraph of Section 3: " less model evaluations." should be " fewer model evaluations."



**Experience Assessment:**

I have published in this field for several years.

**Review Assessment: Checking Correctness Of Derivations And Theory:**

I assessed the sensibility of the derivations and theory.

**Review Assessment: Checking Correctness Of Experiments:**

I assessed the sensibility of the experiments.

**Review Assessment: Thoroughness In Paper Reading:**

I read the paper thoroughly.

---

> ### Author Response · Authors · 2019-11-09
> **Response to Reviewer 4**
>
> We thank the reviewer for the encouraging comments and the suggestions! Please see our answers below.
>
> 1a. Following the suggestion, we changed the paragraphs around Definition 2. We removed the reference to 2D Fused Lasso; our formulation is not the same, and the comment does not help with the presentation. The paragraph following Definition 2 now lists the intuition behind the setup. It now reads:
>
> "Our approach is based on the "high confidence set" approach which has been successful in numerous applications in high dimensional statistics (Candes et al, 2007; Cai et al, 2011; Fan, 2013). The set of $g$ that satisfy the constraint in the formulation is our high confidence set; if $L$ is chosen properly, this set contains the true LEG coefficient, $\gamma(f,x_0,F)$, with high probability. This setup ensures that the distance between $\gamma$ and $\tilde{\gamma}$ is small. When combined with the TV penalty in the objective function, the procedure seeks to find a solution that both belongs to the confidence set and has sparse differences on the grid. Thus, the estimator is extremely effective at recovering $\gamma$ that have small total variation."
>
> 1b. We added a paragraph following the theorem to cite the relevant work and to summarize the proof technique. We now state:
>
> "The proof is built on top of the "high confidence set" approach of Fan (2013). In the proof, we first establish that, for an appropriately chosen value of $L$, $\gamma^*=\gamma(f,x_0,F)$ satisfies the constraint in our formulation with high probability. Then, we make use of TV sparsity of $\tilde{\gamma}$ and $\gamma^*$ to argue that the two quantities cannot be too far away from each other, since both are in the constraint set. The full proof is provided in the Appendix."
>
> 2. We added the time complexity for primal-dual interior point method solvers in our discussion following Definition 2. We note that this complexity is meant to be an upper bound rather than the expected run time. Furthermore, it is possible to obtain much faster rates by utilizing the sparseness of the constraint matrix (Yen et al, 2015) or by using recently proposed stochastic central path methods (Cohen et al, 2018). We leave those as future directions of research. The sample complexity depends on the sparsity of the LEG coefficient, and can be derived from Theorem 1. Ignoring the log terms, the sample complexity is given by $n=O((p_1p_2)^{1/2} s)$. We now state this quantity in the remarks after Theorem 1.
>
> 3a. We would like to ensure that the reviewer is interested in seeing more results like the sensitivity analysis of Section 5.2; with a larger data pool than three randomly selected images, similar to the study in Chen et al (2019). We have just started a new run where we include C-Shapley as a competitor, and will include these results before the end of the rebuttal period.
>
> [Edit on Nov 15: We revised our sensitivity analysis section and now present a new study with 500 images. LEG appears to be at least as good as GradCAM. Please see the new Section 5.2 for the details.]
>
> 3b. According to our understanding, the connectedness of C-Shapley is mainly due to the fact that C-Shapley considers connected pixels while computing the Shapley values, and the connectedness is not explicitly imposed by the procedure.
>
> 3c. Regarding the sampling procedures; C-Shapley goes through all pixels, and computes an approximate Shapley score. For a specific pixel, the procedure returns its marginal contribution, where the contribution is defined as the change of the score of the prediction when that pixel and (a certain combination of) its neighboring pixels are removed (i.e. painted black). The method provides a local estimate of the effect of the pixel. Our approach to the problem is strictly different, as we seek to find a local linear approximation to the function first, and then evaluate the contribution of the pixels based on their contribution to this local linear approximation. Instead of completely removing the pixels, we instead vary their intensity, and try to estimate a smoothed version of the gradient by using such perturbations. In that sense, our method is more closely related to saliency approaches that are based on computing the gradient of the predictor with respect to the input.
>
> References:
> - Yen, Ian En-Hsu, et al. "Sparse linear programming via primal and dual augmented coordinate descent." Advances in Neural Information Processing Systems. 2015.
> - Cohen, Michael B., Yin Tat Lee, and Zhao Song. "Solving linear programs in the current matrix multiplication time." Proceedings of the 51st Annual ACM SIGACT Symposium on Theory of Computing. ACM, 2019.
> - Chen, Jianbo, Le Song, Martin J. Wainwright and Michael I. Jordan. "L-Shapley and C-Shapley: Efficient Model Interpretation for Structured Data". International Conference on Learning Representations. 2019.

---

### Official Review · AnonReviewer5 · 2019-11-02
**Official Blind Review #5**

**Rating:** 8

**Review:**

Summary
This paper proposes an attribution method, linearly estimated gradient (LEG) for deep networks in the image setting.
The paper also introduces a variant of the estimator called LEG-TV, which includes a TV penalty, and provides a
theorem on the convergence rate of the estimator. The paper finds that the LEG attributions pass sanity
checks.

My recommendation
Overall, I am recommending this paper as a weak accept. There are several points to address with
regards to the exposition and flow of the paper, which is my biggest issue with this paper. I believe
the authors can address this point and I am willing to raise my point on this basis. The paper also
provide some theoretical analysis of the proposed method, which is typically lacking for most
of the interpretation methods in this domain.


Possible Improvements
- The LEG method is not sufficiently motivated. Here, I am specifically referring to the functional
form of the estimated itself in definition 1. See the question section  for some of the issues
I raised there.
- From figure 4, we see that the method passes the proposed sanity checks which seem like a
key motivation for this work, however, the authors don't give an explanation for why this is the
case.
- The paper notes that LEG can be estimated using an LP; it would have been great for the authors
to completely spell this out in the appendix or somewhere in the text. What is the exact form of the
LP? What are the constraints?
- As the authors know, the two evaluations presented in the paper: sanity checks, and the zeroing
out procedure (in figure 5) don't actually tell us which method is a good method, just rule out a method.
I would encourage the authors to design a toy task where the ground truth attributions are know, then
train a model to be 100 percent or so accurate on this task. You can then obtain LEG-TV estimates
from this model and compare to the ground-truth.
- I found the proof of lemma 1 confusing, the authors say it follows trivially, but I don't see it. For example,
there should be a factor of 2 somewhere after taking the derivative wrt to $vec(g)$, but I don't see it. It is
fine for the authors to spell out the derivation here if possible.
-The paper ends quite abruptly with no conclusion or discussion. It would be great to include a wrap up
section that puts the contributions into context.
- I get the sense that this method should be computationally intensive, though the paper says otherwise.
It is fine for a method to be computationally intensive, but can the authors speak to this issue?


Some Questions
Definition 1: I had a difficult time understanding this definition. What is $g$ here? I assume
it is the gradient based on the reference to the first order Taylor expansion. In addition, why
is the estimand squared? Further, What does it mean to take expectation wrt $F + x_0$. I was
particularly confused by the last point, because F is a continuous distribution, while presumably
$x_0$ is the point of interest. The paper notes in several places that it can sample from $F + x_0$,
is this equivalent to sampling from $F$ and adding point $x_0$?

What is LEG0 in figure 5?

Is $\kappa$ in your theorem 1, the condition number of the covariance matrix of the perturbation?


Conclusion
Overall, the paper provides a nice method along with analysis on convergence rates and other statistical
properties. Several of the key issues/questions I have about the paper are raised above. None of these
should be dealbreakers but would require the authors to flesh out more details and possibly justify
certain choices. In general, I think more effort should be put into the flow and writing of the paper.
Overall, this is an interesting contribution.


## After reading author responses
I believe the authors have clarified and improved the readability of the paper and clarified several of the questions
that I had. I am raising my score to an accept. While I believe this is a valuable contribution to the sea of attribution methods that have now been published, like the authors noted, it is still not clearly if attribution methods as a whole
are useful of decision making or understanding of a model by either a generic end user or the model developer.
This is a huge problem in this area that deserves significant attention. This said, the goal of this current paper is to
take a step towards developing a principled method, so this is a step in that direction perhaps.

**Experience Assessment:**

I have published one or two papers in this area.

**Review Assessment: Checking Correctness Of Derivations And Theory:**

I assessed the sensibility of the derivations and theory.

**Review Assessment: Checking Correctness Of Experiments:**

I carefully checked the experiments.

**Review Assessment: Thoroughness In Paper Reading:**

I read the paper thoroughly.

---

> ### Author Response · Authors · 2019-11-09
> **Response to Reviewer 5 - Comments**
>
> > "The LEG method is not sufficiently motivated. Here, I am specifically referring to the functional form of the estimated itself in definition 1. See the question section  for some of the issues  I raised there."
>
> - We hope that our added figure (Figure 1) and the captions provide a better picture for our motivation. We are happy to include a more detailed explanation if it is needed.
>
> > "From figure 4, we see that the method passes the proposed sanity checks which seem like a
> key motivation for this work, however, the authors don't give an explanation for why this is the
> case."
>
> - We added the following discussion to the subsection. We now state:
>
> "Our procedure treats the classifier as a black-box and the explanations offered by LEG-TV are based solely on the predictions made by the neural network. During the sanity check, when the weights of the neural network are randomly perturbed, the predictions change significantly and no longer depend on the input. Thus, we expect the local linear approximations of the underlying function to be flat, which would result in saliency scores of zero for all of the pixels. Finally, small artifacts that might arise in this process, such as positive or negative saliency scores with no spatial structure, should be smoothed over due to the TV penalty, further robustifying our procedure."
>
> > "The paper notes that LEG can be estimated using an LP; it would have been great for the authors
> to completely spell this out in the appendix or somewhere in the text. What is the exact form of the
> LP? What are the constraints?"
>
> - We now provide the exact form of the LP in the appendix, under the alternative formulation.
>
> > "As the authors know, the two evaluations presented in the paper: sanity checks, and the zeroing out procedure (in figure 5) don't actually tell us which method is a good method, just rule out a method. I would encourage the authors to design a toy task where the ground truth attributions are know, then train a model to be 100 percent or so accurate on this task. You can then obtain LEG-TV estimates from this model and compare to the ground-truth."
>
> - We also would like to have such a setup for showing our method's efficacy. However, there are major limitations. Most importantly, all of the methods have different estimands. We do not know how to come up with an example where all saliency methods would seek to estimate the same ground-truth. Furthermore, the LEG estimand changes with respect to $\Sigma$, and using different distributions result in different ground-truths; that is, it is hard to compare LEG even to itself! We would be very happy to try out new simulations if the reviewer could suggest possible directions to avoid the multiple estimand issue. Additionally, we argue that the sensitivity analysis is a good indicator of which method performs better. If a method is successful at identifying regions of high saliency, then the probabilities should significantly drop when those regions are masked. Thus, rate of the probability change would be a reliable indicator for understanding which methods overperform.
>
> > "I found the proof of lemma 1 confusing, the authors say it follows trivially, but I don't see it. For example, there should be a factor of 2 somewhere after taking the derivative wrt to , but I don't see it. It is fine for the authors to spell out the derivation here if possible."
>
> - The proof now includes the derivation. Thanks to the comment, we also realized that an implicit assumption (that F has to be centered) was not listed, and we corrected the text.
>
> > "The paper ends quite abruptly with no conclusion or discussion. It would be great to include a wrap up
> section that puts the contributions into context."
>
> - We now include a short discussion that summarizes our contributions.
>
> > "I get the sense that this method should be computationally intensive, though the paper says otherwise. It is fine for a method to be computationally intensive, but can the authors speak to this issue?"
>
> - The method is computationally intensive, but its computational complexity is much lower compared to its alternatives which often rely on non-convex formulations. We now include the time complexity for solving a linear program with an interior point method in our discussion following Definition 2, and also provide the runtime using commercial software (which is roughly 3 seconds on a standard PC).

---

> ### Author Response · Authors · 2019-11-09
> **Response to Reviewer 5 - Questions**
>
> We thank the reviewer for the detailed comments and the constructive criticism! Following the suggestions, we made numerous changes to improve the readability and the flow of the paper. In our first reply, we respond to the questions. We provide our comments to the suggestions in our second reply.
>
> > "What does it mean to take expectation wrt F+x_0. I was particularly confused by the last point, because F is a continuous distribution, while presumably x_0 is the point of interest. The paper notes in several places that it can sample from F+x_0, is this equivalent to sampling from F and adding point x_0?"
>
> - Exactly! We note this in our notation section. We are happy to reiterate the description (or introduce the notation at a better point in text) if the reviewer finds that would improve the flow.
>
> > "Definition 1: I had a difficult time understanding this definition. What is g here? I assume
> it is the gradient based on the reference to the first order Taylor expansion."
>
> - Our formulation is based on a linear approximation of the function $f(\cdot)$ around the point of interest $x_0$. For this approximation, the most sensible approach would be to use the gradient of $f(\cdot)$ at $x_0$; however this results in solutions that are too noisy and not meaningful for interpretation. To avoid such issues, we instead seek to find a new coefficient that can provide a reliable linear approximation around the neighborhood of $x_0$. As the reviewer noted, this definition is motivated by a first order Taylor expansion. In order to improve our presentation, we added a new figure (Figure 1) that demonstrates LEG visually on a toy example. If the reviewer does not find this to be appropriate, we can provide more motivation by citing other relevant work in the saliency literature.
>
> > "In addition, why is the estimand squared?"
>
> - We use squared loss as it results in an analytical solution. We also find it to be a natural choice. A similar square based loss is also used in the LIME paper.
>
> > "What is LEG0 in figure 5?"
>
> - LEG0 is the estimate with a smaller choice of $L$, which results in less sparse solutions. Please see the 7th line of the third paragraph of Section 5.2, where we state "For LEG, we provide two solutions, a sparse solution which corresponds to a larger choice of the penalty parameter $L$ and a noisy solution which is obtained with a smaller choice of $L$, denoted by LEG and LEG0, respectively."
>
> > "Is $\kappa$ in your theorem 1, the condition number of the covariance matrix of the perturbation?"
>
> - $\kappa$ is the constant defined in Assumption 1. It is more related to the minimum eigenvalue of $\Sigma$ than its condition number.

---

### Author Response · Authors · 2019-11-15
**Overview of Revisions**

We would like to again thank the five anonymous reviewers for their valuable input. We made numerous additions and changes to the main body and the numerical results of the paper. We believe that the revisions suggested by the reviewers have greatly improved the presentation of the paper, and the new numerical results better demonstrate the validity to our approach.

The main changes are as follows:
- The sensitivity analysis in Section 5.2 has been completely changed. We now present average of the results from 500 images; previous results used 3 samples.
- Additional discussions have been added to improve the readability of the paper. Mainly:
	- A new figure (Figure 1) has been added to Section 2 to motivate the formulation for LEG.
	- We now state empirical and theoretical run time complexities of the procedure.
	- We describe why the proposed linear program is an ideal choice for estimating LEG.
	- A sketch of the proof technique for Theorem 1 has been added to the main text.
	- The paper now concludes with a discussion section that summarizes our contributions.

Due to the additions, the paper is now a little over 9 pages. We hope that this will not inconvenience the reviewers.

---

### Decision · Program_Chairs · 2019-12-19

**Decision:**

Reject

**Comment:**

This submission proposes a statistically consistent saliency estimation method for visual model explainability.

Strengths:
-The method is novel, interesting, and passes some recently proposed sanity checks for these methods.

Weaknesses:
-The evaluation was flawed in several aspects.
-The readability needed improvement.

After the author feedback period remaining issues were:
-A discussion of two points is missing: (i) why are these models so sensitive to the resolution of the saliency map? How does the performance of LEG change with the resolution (e.g. does it degrade for higher resolution?)? (ii) Figure 6 suggests that SHAP performs best at identifying "pixels that are crucial for the predictions". However, the authors use Figure 7 to argue that LEG is better at identifying salient "pixels that are more likely to be relevant for the prediction". These two observations are contradictory and should be resolved.
-The evaluation is still missing some key details for interpreting the results. For example, how representative are the 3 images chosen in Figure 7? Also, in section 5.1 the authors don't describe how many images are included in their sanity check analysis or how those images were chosen.
-The new discussion section is not actually a discussion section but a conclusion/summary section.

Because of these issues, AC believes that the work is theoretically interesting but has not been sufficiently validated experimentally and does not give the reader sufficient insight into how it works and how it compares to other methods. Note also that the submission is also now more than 9 pages long, which requires that it be held to a higher standard of acceptance.

Reviewers largely agreed with the stated shortcomings but were divided on their significance.
AC shares the recommendation to reject.